# Glass-Forming Ability and Corrosion Resistance of Al_88_Y_8−x_Fe_4+x_ (x = 0, 1, 2 at.%) Alloys

**DOI:** 10.3390/ma14071581

**Published:** 2021-03-24

**Authors:** Rafał Babilas, Monika Spilka, Katarzyna Młynarek, Wojciech Łoński, Dariusz Łukowiec, Adrian Radoń, Mariola Kądziołka-Gaweł, Piotr Gębara

**Affiliations:** 1Department of Engineering Materials and Biomaterials, Silesian University of Technology, Konarskiego 18a, 44-100 Gliwice, Poland; rafal.babilas@polsl.pl (R.B.); monika.spilka@polsl.pl (M.S.); wojciech.lonski@polsl.pl (W.Ł.); dariusz.lukowiec@polsl.pl (D.Ł.); 2Institute of Non-Ferrous Metals, ul. Sowińskiego 5, 44-100 Gliwice, Poland; adrianr@imn.gliwice.pl; 3Institute of Physics, University of Silesia, 75 Pułku Piechoty 1, 41-500 Chorzów, Poland; mariola.kadziolka-gawel@us.edu.pl; 4Department of Physics, Częstochowa University of Technology, Armii Krajowej 19, 42-200 Częstochowa, Poland; piotr.gebara@pcz.pl

**Keywords:** Al-based metallic glasses, X-ray diffraction, transmission electron microscopy, Mössbauer spectroscopy, electrochemical measurements

## Abstract

The effect of iron and yttrium additions on glass forming ability and corrosion resistance of Al_88_Y_8-x_Fe_4+x_ (x = 0, 1, 2 at.%) alloys in the form of ingots and melt-spun ribbons was investigated. The crystalline multiphase structure of ingots and amorphous-crystalline structure of ribbons were examined by a number of analytical techniques including X-ray diffraction, Mössbauer spectroscopy, and transmission electron microscopy. It was confirmed that the higher Fe additions contributed to formation of amorphous structures. The impact of chemical composition and structure of alloys on their corrosion resistance was characterized by electrochemical tests in 3.5% NaCl solution at 25 °C. The identification of the mechanism of chemical reactions taking place during polarization test along with the morphology and internal structure of the surface oxide films generated was performed. It was revealed that the best corrosion resistance was achieved for the Al_88_Y_7_Fe_5_ alloy in the form of ribbon, which exhibited the lowest corrosion current density (j_corr_ = 0.09 μA/cm^2^) and the highest polarization resistance (R_p_ = 96.7 kΩ∙cm^2^).

## 1. Introduction

Aluminum is commonly used for structural applications due to many favorable features such as low density, good mechanical properties, and corrosion resistance, as well as easy recyclability [1]. Amorphous aluminum alloys with ultra-high specific strength and Al content of 80 to 95 at.% were produced by an ultra-fast cooling method at the end of the 20th century. Various studies showed that an amorphous structure can only be obtained by rapid solidification of aluminum alloys with the addition of rare earth (RE) metals (Gd or Ce; 3–20 at.%), yttrium, or one or two transition metals (TM; Fe, Ni, and/or Co; 1–15 at.%) [2]. The mechanical properties of these alloys depend on the local atomic and long-range structural order. For example, the amorphous, single-phase Al_85_Ni_5_Y_10_ alloy achieved a tensile strength value of 1260 MPa. The fcc-Al nanocrystals are precipitated during the transformation of primary amorphous structure in Al-based alloys (>87 at.%). The result is increasing the tensile strength to 1.5 GPa without reducing plasticity [3]. However, partial or complete crystallization often significantly reduces the corrosion resistance [4]. Aluminum alloys with amorphous structure are characterized by good corrosion resistance because of homogeneity and absence of microstructural imperfections [5,6], and their high solubility of corrosion-resistant elements. However, these materials are susceptible to pitting corrosion generated by chloride ions which can cause premature destruction [7].

Bulk nanocrystalline aluminum alloys are attractive structural or coating materials for aerospace and automotive applications, as well as for light-weight machine parts, because of high strength, hardness, corrosion resistance, electrical conductance, and flexibility [8,9]. Their high specific strength and outstanding corrosion resistance also render these materials applicable for the medical field [5]. However, amorphous Al-based alloys with approximately 90 at.% of aluminum content show poor paramagnetic behavior and are considered relatively uninteresting from a magnetic perspective [10]. Addition of alloying elements in Al-based amorphous alloys can improve their glass-forming ability (GFA), thermal stability, strength, ductility, electrochemical and magnetic properties [1,11,12,13,14,15]. For example, the addition of selected alloying elements of transition and rare-earth metals like Ni, Fe, Co, Ce, Y, and/or Gd (≤15 at.%) enhances the GFA. In addition, Co, Ni, or Fe also improve the local corrosion resistance of the Al-based alloys with amorphous structure [16]. Addition of iron has a positive impact on increasing strength and stiffness of aluminum alloys, whereas addition of nickel increases the modulus of elasticity and improves mechanical strength at high temperatures [17]. Higher content of rare-earth and/or transition metal elements has an influence on the thermal stability and phase content of the alloys after crystallization. For example, an increase in yttrium content causes an increase in the primary crystallization temperature of Al-RE-TM amorphous alloys [18].

In the last decade, Al-Fe-Y metallic glasses have been studied precisely with regard to atomic structure and crystallization scheme, as well as the effects of microalloying additions on their glass forming ability [3,14,15,19,20,21,22,23]. However, the research gap is the structural analysis of Al_88_Y_8−x_Fe_4+x_ (x = 0, 1, 2 at.%) alloys with the phase composition: fcc-Al, Al_3_Y, and Al_3_FeY_2_. The aim of this work is to assess the impact of alloying additions (Fe and Y) on amorphous phase formation. Also, very few studies have been dedicated to revealing the corrosion resistance of these materials. Considering that corrosion resistance is extremely important to the practical applications of Al-based metallic glasses as engineering materials, the primary aim of the present work is to investigate the influence of alloying elements on the corrosion behavior of Al-Y-Fe alloys obtained by melt spinning (ribbons) or slow casting methods (ingots). The effect of corrosion process on the sample’s surface and chemical reactions were analyzed.

## 2. Materials and Methods

The studies described herein were provided on Al_88_Y_8−x_Fe_4+x_ (x = 0, 1, 2 at.%) alloys in the form of ingots and ribbons. The ingots were prepared by induction melting for the appropriate mixtures of elements, such as Al (99.9%), Y (99.9%), Fe (99.9%), in Al_2_O_3_ crucible under Ar atmosphere with purity of 99.9%. The external morphology of alloys in the form of ingots after casting was presented in Figure 1. The solidification conditions of the alloys were similar, hence no differences in the morphology of the obtained ingots are visible. The samples in the form of ribbons with a thickness of about 30 µm were prepared by the Bühler Melt Spinner SC apparatus (Edmund Bühler GmbH, Hechingen, Germany) at a wheel speed of 30 m/s.

The structure of ingots and melt-spun alloys was investigated by X-ray diffraction (XRD) and Mössbauer spectroscopy (MS). X-ray diffraction patterns were recorded for samples in a form of powder by 2θ angular range from 10° to 90°. Phase analysis was performed using a Mini Flex 600 (Rigaku, Tokyo, Japan) equipped with a copper tube as an X-ray source and D/TEX strip detector.

Mössbauer ^57^Fe transmission spectra were recorded at room temperature by MS96 spectrometer (Tampa, FL, USA) with a linear ^57^Co: Rh source (25 mCi), multichannel analyzer, absorber, and detector. The spectrometer was calibrated with α-Fe foil. The numerical analyses of the Mössbauer spectra were performed with the WMOSS software (Ion Prisecaru, WMOSS4 Mössbauer Spectral Analysis Software, 2009–2016).

The high-resolution transmission electron microscopy (HRTEM) was used to determine structure and phase analysis by S/TEM TITAN 80–300 (FEI Company, Hillsboro, OR, USA).

The electrochemical corrosion investigations were provided in 3.5% NaCl solution at room temperature using an Autolab 302 N potentiostat (Metrohm AG, Herisau, Switzerland), which was controlled by the NOVA software (version 1.11). The measurements were performed in a cell with a water jacket equipped in a saturated calomel electrode (SCE), a platinum counter electrode, and a working electrode (sample). The corrosion behavior was estimated by collecting the open circuit potential (E_OCP_) variation versus SCE. Corrosion current density (j_corr_) was determined by the Tafel extrapolation using the β_a_ and β_c_ coefficients. The polarization resistance (R_p_) was defined as the slope of a potential versus current density plot.

Microstructure and surface morphology changes of alloys in the form of ingot were observed using a scanning electron microscope (SEM) Supra 35 (Carl Zeiss, Oberkochen, Germany) with the energy-dispersive X-ray spectroscopy (EDX) EDAX. Additionally, based on the XRD results after corrosion the mechanism of chemical reaction was proposed.

## 3. Results

### 3.1. Structural Analysis

The structure of the Al-based alloys in the as-cast state was evaluated by X-ray diffraction. The XRD patterns of Al_88_Y_6_Fe_6_, Al_88_Y_7_Fe_5_, and Al_88_Y_8_Fe_4_ are presented in Figure 2; ingots—(a) and ribbons—(b). Induction melted alloys were characterized by a multi-phase crystalline structure. The phase composition was the same for all alloying compositions. The ingots were consisted of the following phases: α-Al, Al_3_Y, and Al_10_Fe_2_Y. The same phase composition was also confirmed in the literature for Al_80_Fe_10_Y_10_ (induction melted) [24] and Al_82.71_Y_11.22_Fe_6.07_ (prepared by arc-melting) alloys [25]. The XRD patterns of Al_88_Y_8_Fe_4_ and Al_88_Y_7_Fe_5_ ribbons showed broad diffraction lines as well as sharp peaks corresponding to the following crystalline phases: α-Al, Al_3_Y, Al_3_FeY_2_, Al_7_Fe_5_Y, and Al_13_Fe_4_. It was noted, that with increased Fe content, the diffraction peaks related to the crystalline phases disappeared and for Al_88_Y_6_Fe_6_ alloy, the close to pure amorphous phase alloy was obtained. The observed first diffuse maximum (Figure 2b) is associated with the presence of a large number of nanocrystals of Al_3_FeY_2_ phase.

The α-Al, Al_3_Y, and AlFeY phases formed after preannealing amorphous Al_88_Y_7_Fe_5_ alloy at 773K were previously reported by Saksl et al. [3]. In addition, Yin et al. [7] published the XRD pattern of an Al_88_Y_7_Fe_5_ master alloy, which showed diffraction peaks corresponding to α-Al, Al_3_Y, Al_3.25_Fe, and Fe_2_Y phases. Other investigations of the Al_88_Y_5_Fe_7_ amorphous alloy were carried out by Yang et al. [26]. The authors showed that nano-sized aluminum crystals precipitate during isothermal annealing at 280 °C for 30 min. Additionally, isothermal annealing leads to further growth of aluminum crystals and precipitation of intermetallic phases at 370 °C for 30 min [26].

The SEM microscope images of the Al_88_Y_8_Fe_4_, Al_88_Y_7_Fe_5_, and Al_88_Y_6_Fe_6_ ingots are illustrated in Figure 3. The SEM images present the existence of α-Al, Al_3_Y and Al_10_Fe_2_Y phases (identified by EDX). The Al_3_Y phase crystallized in the form of elongated, regular grains. The Al_10_Fe_2_Y formed dendritic structures, which are especially visible for the alloys Al_88_Y_8_Fe_4_ (Figure 3a) and Al_88_Y_7_Fe_5_ (Figure 3b). In the work [24] for the Al_80_Fe_10_Y_10_ alloy, the same phases were identified, however, there was a greater proportion of Al_3_Y and Al_10_Fe_2_Y phases. The paper proposes a crystallization mechanism based on the differential thermal analysis (DTA) cooling curves. First, Al_3_Y phase was crystallized (~1000 °C), then Al_10_Fe_2_Y (~900 °C), and finally aluminum phase (~639 °C).

To identify the detailed structure of Al_88_Y_6_Fe_6_ alloy, the HRTEM examinations were carried out in the ribbon sample in as-cast state (Figure 4). Selected area electron diffraction (SAED), presented in Figure 4b, confirmed the presence of α-Al, Al_13_Fe_4_, and Al_3_FeY_2_ phases in the studied ribbon. These phases were also detected by XRD analysis. To further investigate the structure of as-cast ribbon, the high-resolution image (Figure 4a) was treated by inverse Fourier transfer method (IFT). The HRTEM image with selected areas present a crystalline (Figure 4c,e) and amorphous (Figure 4d) regions. The interplanar spacings in the marked crystalline areas were d = 0.2 nm. The revealed values of d-spacings were found to fit with the interplanar spacings of α-Al phase.

In the works to date, the Al_82_Fe_16_Y_2_, Al_85_Y_10_Fe_5_, Al_88_Y_5_Fe_7_ alloys were described as fully amorphous [27,28]. The occurrence of crystalline phases for these compositions was described as the effect of crystallization during heating [27] or cold-rolling [28] samples from the amorphous state. The publication [29] shows a diagram where the presence of amorphous (ductile and brittle), amorphous-crystalline, and crystalline phases was marked for individual atomic shares of Y and Fe. According to this diagram, the Al_88_Y_8−x_Fe_4+x_ alloys (x = 0, 1, 2 at.%) are located on the border of the amorphous and crystalline regions, which may suggest the coexistence of these structural states [29]. Additionally, Foley and Perepezko described in work [30] that TEM observations showed the presence of large micrometer sized intermetallic phases in the amorphous matrix in Al_88_Y_7_Fe_5_ alloys.

Figure 5 shows room temperature Mössbauer spectra of Al_88_Y_8−x_Fe_4+x_ (x = 0, 1, 2 at.%) alloys in the form of ingots. The measurements show that no magnetic ordering is detected in the investigated alloys, which is not surprising considering the low concentration of Fe atoms and that both Y and Al are non-magnetic. These spectra were fitted with two components, which indicate that iron atoms are situated correspondingly in two different neighborhoods of Al atoms. The hyperfine interaction parameters obtained from the analysis of these spectra are summarized in Table 1. Hyperfine parameters of the doublet, which is the main component of all spectra, correspond to Fe atoms located in Al_10_Fe_2_Y structure [31]. The singlet corresponding to the solid solution of Fe in Al [31] was also found in these spectra. Room temperature Mössbauer spectra of Al_88_Y_8−x_Fe_4+x_ (x = 0, 1, 2 at.%) ribbons in their as-cast states are shown in Figure 6. The hyperfine parameters are summarized in Table 2. All Mössbauer spectra for melt-spun ribbons were fitted using two doublets representing two different local environments of a ^57^Fe nuclide which also means that iron atoms are situated correspondingly in two different neighborhoods of Al and Y atoms. Moreover, the values of isomer shifts of these doublets are quite similar and correspond to amorphous aluminum containing Fe [32,33] not well-crystallized aluminum phase. However, these doublets have significantly different values of quadrupole splitting. Such differences can be the result of substitution of Fe atoms with Y atoms, which have a higher atomic radius, leading to greater distortion in the local environment of an Fe site, and in consequence higher values of quadrupole splitting. For this reason, we assume that the first doublet is connected with aluminum areas rich in Fe and Y atoms and another doublet corresponds to aluminum areas containing only Fe atoms.

### 3.2. Corrosion Properties

The electrochemical results obtained from measurements in 3.5% NaCl solution at 25 °C for Al_88_Y_8−x_Fe_4+x_ (x = 0, 1, 2 at.%) alloys in the form of ingots and ribbons are presented in Figure 7. Table 3 compares quantitative results of electrochemical tests. The corrosion resistance can be assessed on the basis of polarization curves by the extrapolation of the Tafel anode and cathode slopes (β_a_, β_c_) according to the Stern–Geary method. Based on the Equation (1), E_corr_, R_p_, and j_corr_ were determined [34].
(1)$$j_{\mathrm{corr}}=\frac{\beta_{a}\left|\beta_{c}\right|}{2.303\;\left(\beta_{a}+\left|\beta_{c}\right|\right)}\frac{1}{R_{p}}=\frac{B}{R_{p}}$$

The comparison of the values of open circuit potential and corrosion potential is used in assessing corrosion resistance between individual studied alloys under the same conditions. This approach was also described in other publications [35,36].

It is noticed that the E_OCP_ of all samples during the initial immersion time displayed unstable behavior (Figure 7a,c). The highest open circuit potential values after 3600 s were recorded for Al_88_Y_6_Fe_6_ alloys in the form of ingot (−732 mV) and ribbon (−729 mV). The lowest values of E_OCP_ were indicated by the Al_88_Y_7_Fe_5_ alloy in the form of ingot (−1041 mV), while in case of ribbons, Al_88_Y_8_Fe_4_ (−779 mV). Figure 7b,d shows the polarization curves. The most favorable values of the corrosion potential were indicated by the Al_88_Y_6_Fe_6_ alloy for ingot (−707 mV), for ribbon (−648 mV), while the lowest E_corr_ values were observed in the Al_88_Y_7_Fe_5_ alloy for ingot (−987 mV) and ribbon (−691 mV). However, the lowest value of corrosion current density (j_corr_ = 0.09 μA/cm^2^) and the highest polarization resistance (R_p_ = 96.7 kΩ∙cm^2^) were obtained for the Al_88_Y_7_Fe_5_ alloy in the form of ribbon. The discrepancy between the values of E_corr_ and E_OCP_ and i_corr_ indicates a better corrosion resistance of Al_88_Y_7_Fe_5_ alloys, because the potentials are thermodynamic values while the corrosion current density is kinetic and related to the corrosion rate. Despite the higher potentials, the Al_88_Y_8_Fe_4_ and Al_88_Y_6_Fe_6_ alloys will be subject to corrosion phenomena faster.

In all studied compositions, the effect of rapid solidification on the improvement of E_OCP_, E_corr_, j_corr_ values is clear. In case of R_p_, more favorable values occurred just for Al_88_Y_7_Fe_5_ and Al_88_Y_6_Fe_6_ alloys. Similar, the potentiodynamic polarization curves of Al_88_Fe_5_Y_7_ ribbons in 3.5% NaCl solution were obtained in work [7]. More favorable corrosion resistance properties were observed for the samples in the form of ribbons, and these findings are directly related to the structural testing results for these alloys. The ribbon form of the Al_88_Y_6_Fe_6_ alloy has an amorphous-crystalline structure, and, according to the literature [4], single-phase or close to single-phase structural characteristics improve the corrosion resistance of aluminum alloys. Additionally, studies conducted by Lin et al. [37] confirm that, in partially crystallized alloys, the created phases may enhance the diffusion of passive elements of the alloys to form a protective passive layer. With further crystallization, metallurgical defects, such as grain boundaries and grain edges, are formed in the alloy, thus creating local corrosion-prone regions and decreasing the corrosion resistance of completely crystallized samples. For these reasons, the extent of crystallization significantly affects the corrosion resistance of the alloys. It was also discovered that partially crystallized alloys containing different volume fractions of nano α-Al phases exhibit higher corrosion resistance [37]. Finally, in the study presented in [38], it was found that the volume fraction of nanocrystal α-Al plays an important role in governing the electrochemical corrosion properties of Al-Ni-Y alloys, and complete crystallization causes corrosion resistance deterioration.

Additionally, to determine the corrosion mechanism, as well as the corrosion products, the XRD patterns and SEM micrographs with EDS (Energy Dispersive Spectrometer) were collected for the samples after immersion corrosion test in NaCl. The analysis of results for the Al_88_Y_8_Fe_5_ alloy in a form of ingot are presented in Figure 8. As can be seen in XRD patterns collected for the same sample before and after the corrosion test, the intensity of the diffraction peaks related to the existence of Al and Al_10_Fe_2_Y phases decreased, whereas the changes in the intensity of Al_3_Y phase were not observed. Moreover, some peaks related to the nonstoichiometric Al_2.67_O_4_ phase were indicated for the sample after the corrosion test. It can be concluded that the corrosion resistance of the pure Al, as well as phase with Fe and Y, is much lower than Al_3_Y. The existence of corrosion products on the surface of the alloy was additionally confirmed by analysis of SEM micrographs and EDX spectra (Figure 8b). The presence of Al and O confirms the formation of the passive layer, whereas peak related to the Y can be attributed to the Al_3_Y phase with higher corrosion resistance. As was presented in the literature, the addition and formation of phases with Fe increases the corrosion resistance of pure Al [39]. Therefore, firstly, pure Al reacts with electrolyte and form Al^3+^ ions, which in contact with OH^-^ ions precipitate as Al(OH)_3_ and then transform in air into Al_2.67_O_4_ (porous metal oxide described as a defected spinel structure with some cationic vacancies). With decreasing of the concentration of the Al phase, the same reaction occurs in the Al_10_Fe_2_Y phase, however, the Fe atoms also can form the Fe^2+^ ion, however, the low concentration of the Fe in alloy (5 at.%) made it difficult to identify the products of this reaction (which should be stable oxide: Fe_2_O_3_ formed by oxidation of unstable Fe(OH)_2_ [39]). The corrosion process in electrolyte (NaCl) solution can be described by the Reactions (2–7).
(2)$${\mathrm{Al}}_{\left(\mathrm{from}\;\mathrm{Al}\;\mathrm{and}\;\mathrm{Al}10\mathrm{Fe}2Y\right)}\overset{\mathrm{electrolyte}}{\to }{\;\mathrm{Al}}^{3+}+3e^{-}$$
(3)$${\mathrm{Fe}}_{\left(\mathrm{from}\;\mathrm{Al}10\mathrm{Fe}2Y\right)}\overset{\mathrm{electrolyte}}{\to }{\;\mathrm{Fe}}^{2+}+2e^{-}$$
(4)$$6e^{-}+3O_{2}+6H_{2}O\to 6{\mathrm{OH}}^{-}$$
(5)$${\mathrm{Al}}^{3+}+3{\mathrm{OH}}^{-}\to \;\mathrm{Al}{\left(\mathrm{OH}\right)}_{3}$$
(6)$${\mathrm{Fe}}^{2+}+2{\mathrm{OH}}^{-}\to \;\mathrm{Fe}{\left(\mathrm{OH}\right)}_{2}$$
(7)$$4\mathrm{Fe}{\left(\mathrm{OH}\right)}_{2}+O_{2}+\;2H_{2}O\to \;4\mathrm{Fe}{\left(\mathrm{OH}\right)}_{3}$$

Yin et al. [7] studied the corrosion behavior of an Al_88_Fe_5_Y_7_ alloy in 3.5% NaCl at room temperature with SEM and EDS analysis and observed the formation of an oxide film on the surface of the samples. Further testing of the passive layer using X-ray photoelectron spectroscopy (XPS) techniques confirmed the presence of aluminum, iron, and yttrium oxides. In addition, SEM images did not reveal any signs of pitting, which supports the formation of a passive oxide layer on the surface and the observed corrosion resistance of these alloys in the NaCl environment.

## 4. Conclusions

In the article, the structural and anticorrosion properties of Al_88_Y_8−x_Fe_4+x_ alloys (x = 0, 1, 2 at.%) in a form of ingots and melt-spun samples were investigated. The glass-forming ability was increasing with greater iron content, therefore rapidly quenched alloy containing 6 at.% of Fe was characterized by close to the full amorphous structure. The alloy Al_88_Y_7_Fe_5_ in the form of a ribbon was characterized by the lowest corrosion current density and the highest polarization resistance, what indicates the best corrosion resistance.

## Figures and Tables

**Figure 1 materials-14-01581-f001:**
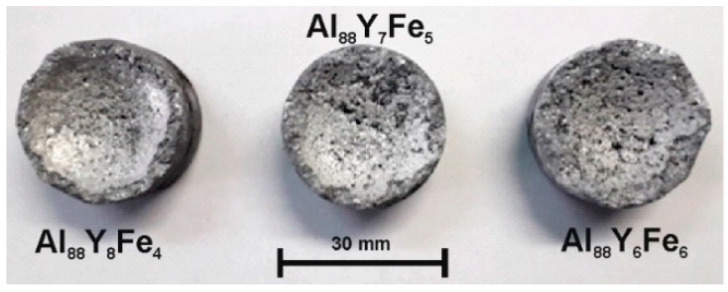
External morphology of Al_88_Y_8_Fe_4_, Al_88_Y_7_Fe_5_, and Al_88_Y_6_Fe_6_ alloys in the form of ingots.

**Figure 2 materials-14-01581-f002:**
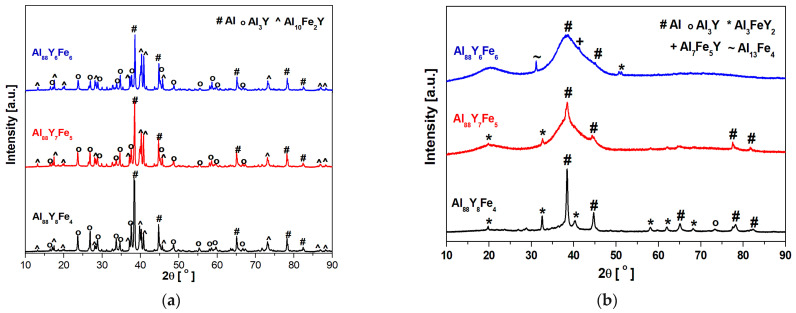
X-ray diffraction (XRD) patterns of Al_88_Y_6_Fe_6_, Al_88_Y_7_Fe_5_, and Al_88_Y_8_Fe_4_ alloys in the form of ingots (**a**) and in the form of ribbons (**b**) in as-cast states.

**Figure 3 materials-14-01581-f003:**
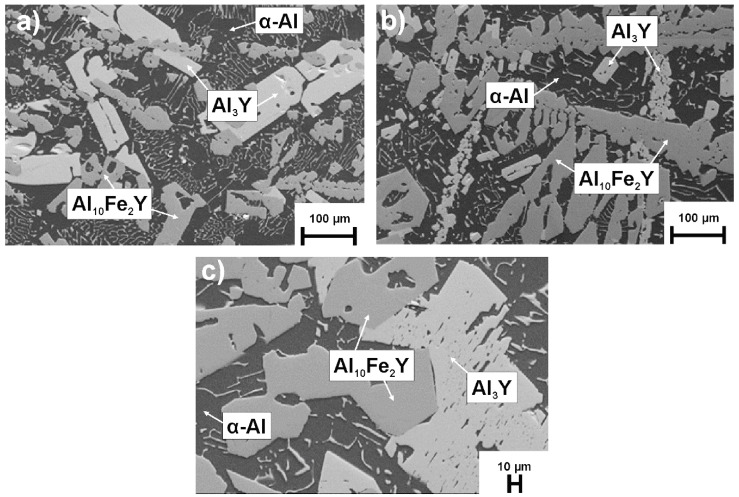
Microstructures of (**a**) Al_88_Y_8_Fe_4_, (**b**) Al_88_Y_7_Fe_5_, (**c**) Al_88_Y_6_Fe_6_ alloys in the form of ingots.

**Figure 4 materials-14-01581-f004:**
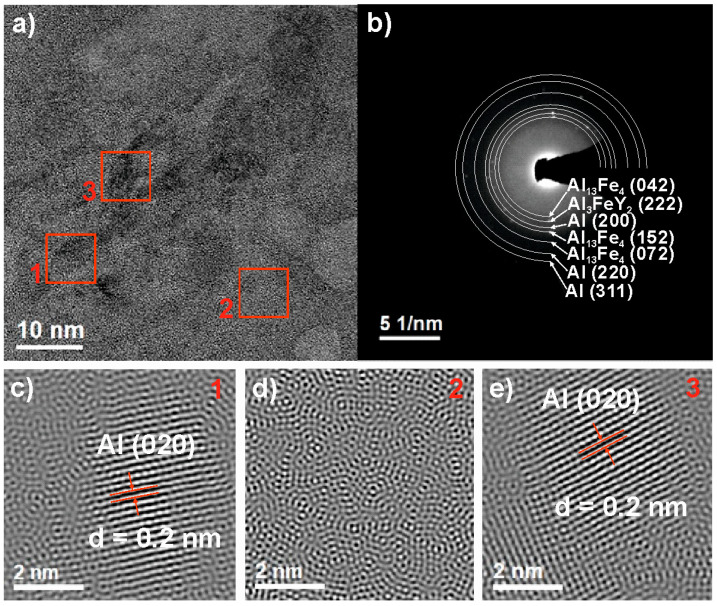
High-resolution transmission electron microscopy (HRTEM) image (**a**), selected area electron diffraction (SAED) pattern (**b**) of Al_88_Y_6_Fe_6_ ribbon, and inverse Fourier transfer (IFT) images (**c**–**e**) from selected areas 1–3.

**Figure 5 materials-14-01581-f005:**
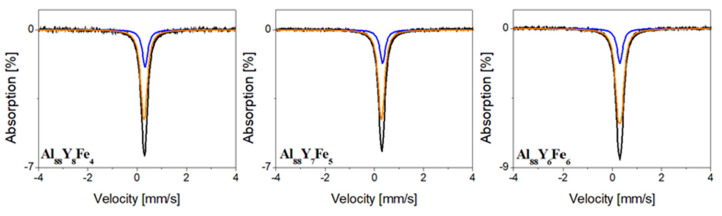
Mössbauer spectra of Al_88_Y_8_Fe_4_, Al_88_Y_7_Fe_5_, and Al_88_Y_6_Fe_6_ alloys in the form of ingots.

**Figure 6 materials-14-01581-f006:**
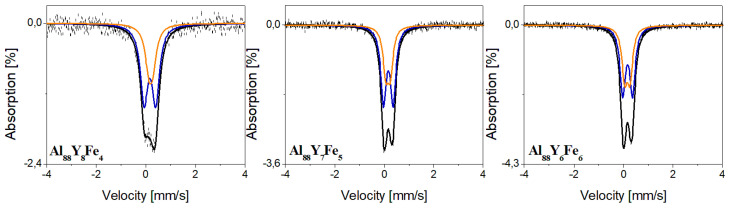
Mössbauer spectra of Al_88_Y_8_Fe_4_, Al_88_Y_7_Fe_5_, and Al_88_Y_6_Fe_6_ alloys in the form of ribbons.

**Figure 7 materials-14-01581-f007:**
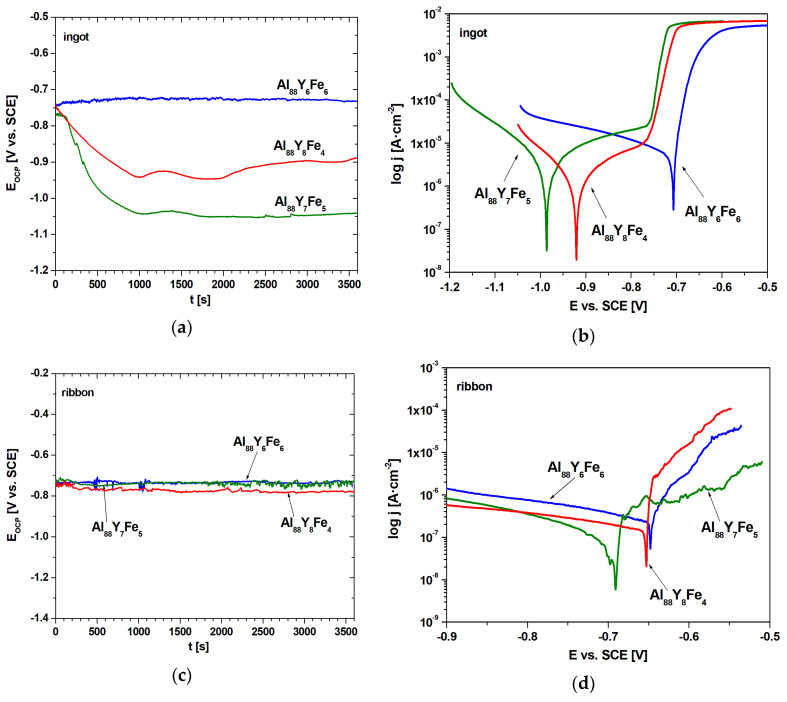
Changes of the open-circuit potential with time (**a**,**c**) and polarization curves (**b**,**d**) in 3.5% NaCl solution at 25 °C of ingots (**a**,**b**) and ribbons (**c**,**d**).

**Figure 8 materials-14-01581-f008:**
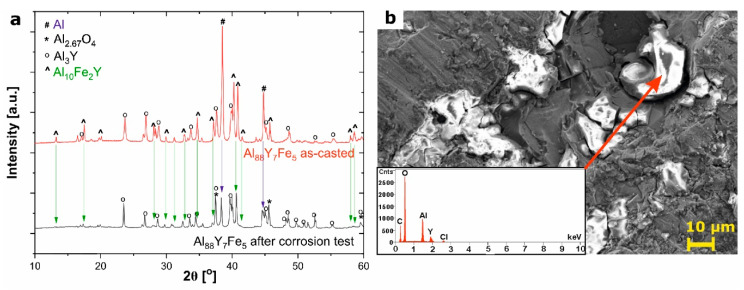
Analysis of the formation process of corrosion products on the surface of Al_88_Y_7_Fe_5_ alloy: (**a**) comparison of XRD patterns of alloy before and after corrosion test with marked identified phases, (**b**) scanning electron microscopy (SEM) micrograph of Al_88_Y_7_Fe_5_ surface after corrosion test (inset shows the energy-dispersive X-ray (EDX) spectrum).

**Table 1 materials-14-01581-t001:** Hyperfine parameters of Al_88_Y_8-x_Fe_4+x_ (x = 0, 1, 2 at.%) alloys in ingot states (IS = isomer shift, QS = quadrupole splitting, FWHM = full width at half maximum, A = relative area derived from the spectra).

Component	IS (mm/s)	QS (mm/s)	FWHM (mm/s)	A (%)	Compound
**Al_88_Y_6_Fe_6_**					
Singlet	0.31	-	0.27	22	Al(Fe)
Doublet	0.30	0.15		78	Al_10_Fe_2_Y [32]
**Al_88_Y_7_Fe_5_**					
Singlet	0.34	-	0.27	22	Al(Fe)
Doublet	0.29	0.09		78	Al_10_Fe_2_Y [32]
**Al_88_Y_8_Fe_4_**					
Singlet	0.34	-	0.27	26	Al(Fe)
Doublet	0.29	0.10		74	Al_10_Fe_2_Y [32]

**Table 2 materials-14-01581-t002:** Hyperfine parameters of Al_88_Y_8−x_Fe_4+x_ (x = 0, 1, 2 at.%) alloys in ribbon states (IS = isomer shift, QS = quadrupole splitting, FWHM = full width at half maximum, A = relative area derived from the spectra).

Component	IS (mm/s)	QS (mm/s)	FWHM (mm/s)	A (%)	Compound
**Al_88_Y_6_Fe_6_**					
Doublet 1	0.17	0.40	0.27	58	Al(Fe,Y)
Doublet 2	0.15	0.22		42	Al(Fe)
**Al_88_Y_7_Fe_5_**					
Doublet 1	0.16	0.39	0.27	63	Al(Fe,Y)
Doublet 2	0.15	0.18		37	Al(Fe)
**Al_88_Y_8_Fe_4_**					
Doublet 1	0.17	0.47	0.36	67	Al(Fe,Y)
Doublet 2	0.20	0.16		33	Al(Fe)

**Table 3 materials-14-01581-t003:** Results of polarization tests of Al-Y-Fe alloys in the form of ingots and ribbons (E_OCP_ = open circuit potential, E_corr_ = corrosion potential, β_a_, β_c_ = anodic and cathodic Tafel slopes, R_p_ = polarization resistance, j_corr_ = corrosion current density).

Alloy	E_OCP_ (Mv)	E_corr_ (mV)	|β_a_| (mV/dec)	|β_c_| (mV/dec)	R_p_ (kΩ∙cm^2^)	j_corr_ (μA/cm^2^)
**Ingots**						
Al_88_Y_8_Fe_4_	−888	−921	167	219	15.2	2.70
Al_88_Y_7_Fe_5_	−1041	−987	62	87	7.9	1.99
Al_88_Y_6_Fe_6_	−732	−707	76	10	1.4	2.67
**Ribbons**						
Al_88_Y_8_Fe_4_	−779	−653	242	4	12.2	0.14
Al_88_Y_7_Fe_5_	−731	−691	197	23	96.7	0.09
Al_88_Y_6_Fe_6_	−729	−648	7	11	14.9	0.12

## Data Availability

Data sharing is not applicable to this article.

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
