# Peer review of "Glass-Forming Ability and Corrosion Resistance of Al88Y8−xFe4+x (x = 0, 1, 2 at.%) Alloys"

_materials, 2021, doi:10.3390/ma14071581_

Round 1

Reviewer 1 Report

The article is devoted to the study of the structure, corrosion properties and magnetic properties of alloys Al88Y8-xFe4+x after slow cooling and in the form of rapidly quenched ribbons. Despite significant changes in work, they did not touch on the most important thing - the essence of the study. In this form, the article is three unfinished independent work. At the same time, the need to measure the magnetic properties of aluminum alloys remains unclear. In the introduction, the authors discuss the use of aluminum alloys as structural or coating mateials.It is not clear how the magnetic properties of the paramagnet are related to this application.

In addition to the remarks indicated above, the following weak points in the work can be distinguished:

  1. Analysis of the microstructure of the alloy is not thoroughly carried out. It is necessary to indicate what is the reason for the appearance of a diffuse maximum at angles ~ 20 degrees (Fig.2b). Microstructure analysis in Figure 3 is completely missed. Differences in observed microstructures are not discussed and structural component analysis is not performed. It is not clear why the authors describe the Mösbauer spectra of partially crystalline alloys with only two doublets related to intermetallic phases. The authors believe that iron atoms are not contained in the amorphous phase? In my opinion, this requires experimental confirmation.
  2. The authors talk about the influence of the formed hydroxides on the formation of oxides on the surface, but it is not clear what is the effect the chlorine ions contained in the solution.
  3. It is not clear what result the authors expected to obtain when studying the magnetic properties.

Despite the considerable efforts of the authors to revise the article, I believe that before the publication of the work, it is necessary to more clearly formulate the purpose of the work and plan the experiment based on the goal, but not on the available equipment.

Reviewer 2 Report

In this paper the authors present the corrosion behaviour of the different Al-based alloys. They also add a section concerning magnetic properties at room temperature of these alloys, which are paramagnetic. The paper needs major revision and additional experiments.

  • Line 30 what is magnetic resistance?
  • Authors compare the ingots, obviously crystalline, with the melt spun ribbons, which are partially amorphous partially crystalline. Although they claim that this kind of microstructure is beneficial for the properties, this happens when it is the result of a controlled crystallization. In order to produce the ribbons, they used a linear speed of 30 m/s, but they could have increased the speed to try to obtain amorphous alloys, 40 and 50 m/s are quite usual in this field. I think this should be a mandatory experiment.
  • Line 136: Authors have to compare the phases identified in the ingots by XRD with the predicted by the ternary phase diagrams.
  • Figure 1. The ingots seem to have a large porosity from the picture. Is this the case? Why?
  • Figure 3: The different phases should be indicated on the micrographies.
  • Line 175-176. Change the sentence for better clarity.
  • Line 177. Which is this AlmFe phase? Is it a solid solution?
  • Line 198: typo.
  • Table 4. The authors have obtained coercive field of about 0.008 T, with the VSM, which for me is quite hard to believe this resolution.

Reviewer 3 Report

It is a good piece of work. Yet, the authors did not reflect some of the previous comments in this manuscript. It would be much better and informative if the manuscript was more carefully written.

There are several issues to be clarified with the presentation and interpretation of the results. Therefore, the reviewer does not feel this paper is ready for publication without revision and corrections. Furthermore, some paragraphs are not well organized. The reviewer recommends getting proof-readings from other colleagues working in the field prior to their submittal.

Comments (with some previous comments that are not sufficiently explained.)

  1. Title: Is it possible to say, “effect of Y addition on” instead of “effect of Fe addition on”? The reviewer doesn’t believe that “effect of T” is more suitable. Just wants to point out that it would be better to change the title. Especially, “Fe addition” doesn’t fit well with “of XX-XX-Fe”.
  2. Abstract: The abstract needs to be informative, structured, concise, and coherent. Explaining what the main finding reveals in comparison to the previous works or how the results add to previous knowledge can be also very informative.
  3. References: Please read /check the references more carefully. For example, line 140
    “The XRD analysis carried out by Horimura et al. [24] showed that the as-quenched Al90Fe5Y5 alloy is composed of amorphous and fcc-Al solid solution, and the grain size of the Al phase 141 is about 5 nm”, the authors of [24] did NOT say that the grain size of the Al90Fe5Y5 alloy is 5 nm. They did mention that rapidly solidified Al90Fe5Y5 is amorphous. This is not the only one.
  4. Fig Caption: Please correct some figure captions which are not clear. For example, Fig 3. “Structure of” to “Microstructures of” or “optical microstructures of”.
  5. Fig3. Please use arrows or labels so that readers can easily read the micrographs.
  6. Please correct stylistic errors.

Again, the results and the contents are meaningful, which can be published somewhere. The major problems are i) poor interpretations of the findings and the overall flow in the main text and ii) many errors that the readers may find everywhere. These points need to be further improved to warrant publication.

Reviewer 4 Report

The author revised the manuscript according to comments from the reviewers. XRD patterns was modified, and the calculation detail of polarization resistance was also described. So I recommend receiving this article.

Round 2

Reviewer 1 Report

Despite the fact that the article has been significantly revised, there are quite a few errors and unresolved issues in it, here are some of them: 

  1. The description of the microstructure of the alloys is completely incorrect.Authors should understand it in more detail.And also explain what the authors understand under the term "Al matrix", and understand in the area of primary crystallization which phase their alloys are in.The authors should also understand the phase composition of the eutectic.
  2. Authors should provide reference to works in which two diffuse maxima are also observed on X-ray diffraction patterns, as well as to works in which the formation of amorphous clusters of different types, with such different interatomic distances, was observed.
  3. Lines 236-255. It is not clear from the text of the article what state of the alloy Al88Y8Fe5 is being discussed. Authors need to clarify this.
  4. For alloys in the form of ingots, a passivation region is observed on the potentiodynamic curves, while such a region is not observed for a ribbon samples.Authors should comment on this feature

The article should be revised in accordance with the comments 

Reviewer 2 Report

In my opinion the paper has been improved. I still have two comments:

  • It would be a good idea to provied TEM analysis of the ribbons to help with the XRD and the TMS interpretation.
  • Line 51-52. The sentence has no verb. It should be re-written.

Reviewer 3 Report

Appreciate your efforts. It reads better than the previous version. Yet there is a lot of room to improve. 
The reviewer believes that it would be better to elaborate further on the manuscript.
Please have a close look at the manuscript and revise it before publication.
The reviewer will leave "accept" since the results are worth to be published.

Round 3

Reviewer 1 Report

The authors have significantly revised the article and undoubtedly improved its quality. However, the authors have not been able to fully understand the observed structure:

  1. In the references presented by the authors to studies on the structure of metallic glasses, it is not indicated that clusters with such a small radius of the coordination sphere can be observed in these metallic glasses. The authors should more carefully analyze the results of X-ray diffraction and establish whether the observed diffuse maximum at small angles is a measurement artifact.
  2. The analysis of the microstructure of the ingots in Figure 3 is still not correct from the point of view of the phase rule and the theory of crystallization. In lines 153-157, the authors refer to their own work on the analysis of the structure, and claim that they observe successive crystallization from a liquid of intermetallic compounds, and then a solid solution of aluminum. I believe that the authors should describe in more detail such a crystallization process, which goes beyond the classical theory of crystallization of ternary alloys. And also indicate how the formation of a double eutectic (Al+Al10Fe2Y) is possible during the primary crystallization of the intermetallic compound Al3Y. 

Round 4

Reviewer 1 Report

  1. The authors believe that the presence of the second diffuse maximum in the X-ray diffraction patterns is associated with phase separation.However, phase separation can be easily observed in TEM.At the same time, when studying the microstructure of an alloy in TEM, the authors observe nanocrystals rather than phase separation.Most likely, the observed first diffuse maximum is associated with the presence of a large number of Al3FeY2 phase (and others) nanocrystals.The authors should consider the obtained results once again.
  2. The investigated alloys are in the region of the ternary eutectic (Al + Al3Y + Al10Fe2Y) of the phase diagram. The mention of the peretectic reaction is fundamentally incorrect, since there is no residual Al2Y phase in the structure after nonequilibrium crystallization, and its formation is not possible in the studied alloys. The occurrence of these peretectic reactions during non-equilibrium crystallization raises great doubts. Moreover, the observed structure of the alloys does not correspond to the structure in the peritectic reaction. I believe that the authors need to carefully study the question of the phase composition of alloys using SEM with EDX

Author Response

This manuscript is a resubmission of an earlier submission. The following is a list of the peer review reports and author responses from that submission.

Round 1

Reviewer 1 Report

This work demonstrated the effect of Fe addition on selected magnetic properties and corrosion resistance of Al-Y-Fe amorphous-crystalline alloys. The findings of the paper are interesting and a lot of detailed effort has been put into it. I have some comments that I hope will improve the quality of the article even further.

  1. The XRD angular described in the Materials and Methods section is inconsistent with the result of Fig. 1.
  2. Why do the XRD patterns of Al-Y-Fe Alloy in the form of ribbon and plate in as-cast state different?
  3. Why did the author mention the work in reference 7 in line 166? Are the results of electrochemical tests, such as corrosion potential and polarization resistance, in this manuscript consistent with those in reference 7?
  4. The author must explain how the polarization resistance and corrosion current density were obtained.
  5. It is difficult to tell whether the alloy has pitted or not because of the cover-up of corrosion products.

Reviewer 2 Report

The result is interesting. The present submission focused on mechanical, magnetic, and corrosion properties Al-Y-Fe system. By analyzing the properties, the investigators were able to provide a guideline and insights for other researchers. It is a reasonable study, of particular interest to metallic glasses and amorphous/crystalline alloys which has been pursued in fields.

However, there are several critical issues to be clarified with the presentation and interpretation of the results. Therefore, I do not feel this paper is ready for publication without revision and corrections. That said, I am concerned with several issues that are not sufficiently addressed in the current manuscript. Furthermore, some paragraphs are not well organized. The reviewer recommends getting proofreadings from other colleagues working in the field prior to their submittal.

  1. Title: “Effect of Fe addition on selected magnetic properties and corrosion resistance of Al-Y-Fe amorphous-crystalline alloys”. It would be better to change the title. Especially, “Fe addition” doesn’t fit well with “of XX-XX-Fe”.
  2. Abstract: The abstract needs to be informative, structured, concise, and coherent. For example, from this sentence, “corrosion resistance of -- alloys are related to changes in the concentrations of alloying elements”, readers may want to know what the tendency is or how sensitive it is, not the fact that the corrosion properties are related to the concentrations. Explaining what the main finding reveals in comparison to the previous works or how the results add to previous knowledge can be also very informative.
  3. Unit: Please check the units. Corrosion density- uAcm2 (Isn’t it uA/cm2 ? )
  4. Materials and Methods: "the plates --- were prepared by re-melting the ingots and then pressure casting" The experimental part is not clearly described in the manuscript. Can you please provide more detail for pressure casting? This section should provide sufficient detail for other scientists to reproduce the experiments presented.
  5. References: Please double check the ref. the number before submitting it. Probably an error might occur during writing. There are two No 1’s.
  6. Results: XRD. The reviewer understands how hard it is when multiple crystalline peaks are overlapped in an XRD pattern. Yet, the indexing needs to be done carefully with detailed info (i.e. jcpds number or space group/lattice parameter) for published materials. It doesn’t seem correct. 1) difficult to identify the structure by using one peak (e.g. Fig1b “# ~ 73 degree”) 2) Contradicted itself (e.g. Fig 1c and 1d - peak ~ 36 : indexed as * Al3FeY2 for Al88Y7Fe5, while # YFe4Al8 for Al88Y6Fe6. Peak ~ 52 : indexed as + Al5Y2 for Al88Y7Fe5 yet > Al3Y for Al88Y6Fe6)
  7. Please read the references more carefully. For example…
    -Example 1 (page 3 line 113) from the manuscript, “The structure of the as-quenched  Al90Fe5Y5  alloys is composed of amorphous, amorphous and fcc-Al solid solution and the grain size of the Al phase is about 5 nm.” vs from the paper “the microstructure of the rapidly solidified sample is amorphous for Al90Fe5Y5” (Original in page109 from the ref. “急冷凝固組織はAl90Fe5Y5ではアモルファス”. This manuscript is written in Japanese. Luckily, I learned it 20 years ago).
    -Example 2  (page 3 line 120) from the manuscript, “Al88Y7Fe5 amorphous alloy carried out by Yang et al. showed that nano-sized aluminum crystals precipitate after isothermal annealing at 280 °C for 30 min” vs from the ref. the paper “The nanosized aluminum crystals precipitate when the isothermal annealing of Al88Y5Fe7 amorphous alloy is performed at 280 C for 30 min.”. Wrong composition. Not Al88Y7Fe5 but Al88Y5Fe7.
  1. Please provide error bars in corrosion data.
  2. In general, discussion parts of each experiment are not well organized. Some of the discussions read like a list of studies from other researchers. It would be better to digest the arguments from pieces of literature and write with the author's voice to explain the data.
  3. Fig4: Is it possible to get higher mag SEM images or TEM? The corrosion may happen very selectively only for some phases. A combination with SEM (SE, BSE) and EDS mapping will be helpful to understand the corrosion mechanism behind.
  4. Page7. XPS: The authors mentioned about their results obtained from XPS. Please provide the XPS data.
  5. Can you provide a broader perspective, readily comprehensible to a scientist? A few paragraphs or sentences which may give a guideline to the following researchers will be very beneficial.

Overall, the reviewer believes that the key experimental results are interesting. However, as argued in the review, some of the data are not sufficient with the presentation and interpretation. I do not, therefore believe that the results and discussions are clean to warrant publication. This work is not ready for publication at this point of time, without revision.

Reviewer 3 Report

The Al-RE-TM alloys have been studied for many years, including the corrosion properties. The authors should make clear why they did the work presented in this manuscript. What distinguishes this work from previous work? Why did the authors expect these alloys to present interesting magnetic properties since the iron content was so low?

The lettering on figures should be increased in size since it is sometimes difficult to read the subscripts for the various alloy compositions. 

Reviewer 4 Report

In this manuscript authors study the effect of Fe alloying on different physical properties of Al-Y-Fe partially amorphous alloys.

Under my opinion in this manuscript neither new knowledge about Al-alloys nor an interesting material from the point of view of technological applications is presented.  In fact, in the manuscript no foreseen potential applications are mentioned.

In the introduction states that aluminum alloys can be used as structural materials. However, in the manuscript the mechanical properties’ results and discussion are very limited. On the other hand, a magnetic hysteresis curve is shown. I wonder why authors consider that studying the magnetization of this alloy at room temperature, yet it is well known that Al-alloys are no ferromagnetic at RT.  

Reviewer 5 Report

The effect of yttrium-iron substitution in Al-Y-Fe alloys on the structure and properties is considered. In the work, measurements of the mechanical, corrosion and magnetic properties of the studied alloys are carried out. There are a large number of unexplained results that are not interconnected with each other. Almost completely no discussion of the results. I do not recommend this article for publication aе current state.

Some remarks on which the basics are my verdict:

  1. Almost all physical, mechanical and corrosion properties depend on the structure of the material.In this work, alloys are obtained both in the form of ribbons and in a bulkThe structure of all alloys varies greatly.This fact has not been discussed by the authors in any way and is not related to the results obtained.
  2. The authors need to explain how 7 or 8 phases can exist simultaneously in a ternary system. (Figure 1 b,c,d)
  3. Mössbauer sspectroscopic results do not match the results of x-ray analysis. Authors need to explain this inconsistency.
  4. In the analysis of XPS spectroscopy, a large amount of carbon was found, the nature of its appearance is not clear from the work
  5. The meaning of conducting magnetic measurements, as well as focusing on them in the title, is not clear from the work
  6. It is not clear what is associated with such a large difference in the microhardness of the alloys in the crystalline state. The authors also compare their results with those of other authors.However, they do not comment on the structural differences between them.